# Coordination between terminal variation of the viral genome and insect microRNAs regulates rice stripe virus replication in insect vectors

Wan Zhao[1,2], Jinting Yu[1,2], Feng Jiang[2,3], Wei Wang[1], Le Kang[1,2,3], Feng Cui[1,2]*

1 State Key Laboratory of Integrated Management of Pest Insects and Rodents, Institute of Zoology, Chinese Academy of Sciences, Beijing, China, 2 CAS Center for Excellence in Biotic Interactions, University of Chinese Academy of Sciences, Beijing, China, 3 Beijing Institutes of Life Science, Chinese Academy of Sciences, Beijing, China

* cuif@ioz.ac.cn

**Data Availability Statement:** All relevant data are within the manuscript and its Supporting Information files.

## Abstract

Maintenance of a balance between the levels of viral replication and selective pressure from the immune systems of insect vectors is one of the prerequisites for efficient transmission of insect-borne propagative phytoviruses. The mechanism regulating the adaptation of RNA viruses to insect vectors by genomic variation remains unknown. Our previous study demonstrated an extension of the 3'-untranslated terminal region (UTR) of two genomic segments of rice stripe virus (RSV). In the present study, a reverse genetic system for RSV in human cells and an insect vector, the small brown planthopper *Laodelphax striatellus*, was used to demonstrate that the 3'-terminal extensions suppressed viral replication in vector insects by inhibiting promoter activity due to structural interference with the panhandle structure formed by viral 3'- and 5'-UTRs. The extension sequence in the viral RNA1 segment was targeted by an endogenous insect microRNA, miR-263a, which decreased the inhibitory effect of the extension sequence on viral promoter activity. Surprisingly, the expression of miR-263a was negatively regulated by RSV infection. This elaborate coordination between terminal variation of the viral genome and endogenous insect microRNAs controls RSV replication in planthopper, thus reflecting a distinct strategy of adaptation of phytoviruses to insect vectors.

## Author summary

Mutations frequently happen when insect-transmitted RNA viruses circulate between insect vectors and plant or mammalian hosts. However, the significance of these mutations for viral fitness in the two distinct organisms is poorly understood. We discovered that a high proportion of rice stripe virus (RSV) had terminally extended genomes when the virus infected insect vectors. In the present study, we found that the extension sequence suppressed viral replication in insect vectors by impairing a special structure

**Funding:** This work was supported by grants from the National Natural Science Foundation of China (No. 32090012) for W.Z., the State Key Research Development Program of China (No. 2019YFC1200504) for W.W., the Chinese Academy of Sciences (No. ZDBS-LY-SM027) for F.C., and the Youth Innovation Promotion Association, CAS (No. 2019086) for W.Z. The funders had no role in study design, data collection and analysis, decision to publish, or preparation of the manuscript.

**Competing interests:** The authors have declared that no competing interests exist.

formed by the two ends of the viral genomes. An endogenous insect small RNA was able to bind the extension sequence to relieve the inhibitory effect. However, the expression of this small RNA was reduced in the presence of RSV to ultimately maintain the inhibitory effect of the extension sequence. This elaborate coordination between virus and vector enables a limited level of RSV replication that does not produce serious damage to vectors, thus reflecting a distinct strategy of adaptation of insect-transmitted plant viruses.

## Introduction

Insect-borne propagative phytoviruses and arboviruses usually induce serious or fatal diseases in plant and mammalian hosts in contrast to the asymptomatic infection in insect vectors [1,2]. Maintenance of a balance between the levels of viral replication and selective pressure from the immune systems of a host or a vector is one of the major factors that determine the distinct outcome of the infection. Genetic variation is frequently observed in insect-borne RNA viruses during viral circulation in two distinct organisms. In addition to nucleotide deletion, insertion, and substitution in the coding regions, various mutations appear in the untranslated terminal regions (UTRs) of the viral genomes [3,4]. The mechanism of the regulation of adaptation of a virus to insect vectors by variations in the UTRs is currently unknown.

The UTRs of single-stranded RNA viruses play the key roles in viral replication and gene transcription. The 3'- and 5'-UTRs of negative-strand RNA viruses form a panhandle structure via distal complementary 15–19 nucleotides (nt) to provide a binding site for viral RNA-dependent RNA polymerase (RdRp) [5,6]. The 3'-UTRs of certain negative-strand RNA viruses, such as influenza A virus, contain the core promoter elements [5]. Mutations in the 3'-UTRs usually negatively impact viral replication, transcription, or translation [7]. Endogenous microRNAs (miRNAs) of the host cells have been frequently reported to target the 3'-UTRs of RNA viruses to regulate viral replication [8,9].

Our previous study demonstrated 16 and 15 nt extensions in the 3'-UTRs of two genomic segments, negative-sense RNA1 and ambisense RNA2, of rice stripe virus (RSV). RSV is a single-stranded RNA virus of the *Tenuivirus* genus [10] efficiently transmitted by small brown planthopper, *Laodelphax striatellus*, in a persistent-propagative manner. RSV causes serious destructive rice diseases in Asian countries [11]. The genome of RSV consists of four RNA segments encoding a nucleocapsid protein (NP), an RdRp, and five nonstructural proteins [12,13]. Similar to other negative-strand RNA viruses, the 5'- and 3'-terminal sequences of each genomic RNA of RSV have sufficient complementarity to form a panhandle structure [14,15]. We have demonstrated that the relative ratios of 3'-terminally extended types of genomic RNA1 and RNA2 increased from around 7% to 35% or 49% with viral replication in planthopper and decreased to less than 3% in the late-stage of infection in the plant hosts, and the extension sequences were deleterious for viral replication in the plants [10]. However, the mechanism and biological significance of the maintenance of the 3'-terminally extended RSV genomes in insect vectors are unknown.

This study investigated the influence of 3'-terminal extensions of the viral genome on the viral promoter activity using a reverse genetic system for RSV in human cells and in small brown planthopper. The adverse effect of the 3'-extension sequences regulates viral replication level in planthopper. However, the extension sequence is responsible for targeting of the viral genome by a conserved insect miRNA.

## Results

### The 3' extensions suppress viral replication in planthopper

We have previously demonstrated that insect-derived RSV have considerably higher ratios of RNA1 and RNA2 with the extended 3'-UTRs (3'-EUTRs) than those in plant-derived RSV, and the 3' extensions inhibit viral replication in the plant hosts [10]. To determine whether the 3' extensions play a similar function in insect vectors, nonviruliferous planthoppers were fed an artificial diet containing equal amounts of viral RNAs derived from viruliferous planthoppers or from rice. Higher levels of viral RNAs were detected in the original crude extracts from insects than in those from rice (Fig 1A), and significantly higher ratios of RNA1 or RNA2 with the 3'-EUTRs were detected in insect-derived RSV than those in plant-derived RSV (Fig 1B). After dilution at various dilution factors and mixing with the artificial diet, the diluted viral crude extracts contained equal amounts of insect- and plant-derived viral RNAs (Fig 1C); however, the ratios of RNA1 or RNA2 with the 3'-EUTRs were no longer different between RSVs derived from two sources (Fig 1D) due to low virus titer. The accumulation of RNA1 and RNA2 in planthopper was assayed by quantitative real-time PCR (qPCR) at 2 h, 1 d, 2 d, 5 d, and 8 d after the insects were fed the artificial diet containing equal amounts of insect- or plant-derived viral RNAs. Less than 2 d after administration, similar RNA levels of RNA1 or RNA2 were detected in the two groups of planthoppers (Fig 1E and 1F). Starting from 2 d after inoculation (DAI), the amounts of RNA1 and RNA2 in planthoppers infected with insect-derived RSV were always lower than those in planthoppers infected with plant-derived RSV corresponding to the higher ratios of RNA1 or RNA2 with the 3'-EUTRs in planthoppers infected with insect-derived RSV (Fig 1E and 1F). This result demonstrated that the 3' extensions are detrimental to viral replication in insect vectors.

### The 3' extensions inhibit viral promoter activity in vitro

Prediction of the secondary RNA structure suggested that the 3' extensions modify a panhandle structure formed by the 3'- and 5'-UTRs of viral RNA1 or RNA2 (S1 Fig) and may thus influence the promoter activity. To verify this hypothesis, we used the pPol I plasmid to generate a reverse genetic system for RSV in human embryonic kidney 293T cells due to lack of planthopper cell lines. The pPol I plasmid is driven by the human RNA polymerase I (Pol I), which normally directs the synthesis of ribosomal RNAs (rRNAs) [16]. The complementary sequence of *Renilla* luciferase (*Rluc*) flanked by the 3'- and 5'-UTRs of RSV RNA1 or RNA2 were inserted downstream of the Pol I promoter to construct the wild-type plasmids named P1-WT for RNA1 and P2-WT for RNA2. The diagrams of the recombinant pPol I plasmids are presented in Fig 2A. RSV *NP* and *RdRp* were incorporated into the pcDNA plasmid to generate the recombinant pcDNA-NP and pcDNA-RdRp plasmids, which express NP and RdRp, respectively. When the P1-WT or P2-WT plasmids were cotransfected with pcDNA-RdRp or pcDNA-RdRp and pcDNA-NP into 293T cells, the *Rluc* reporter RNA containing viral 3'- and 5'-UTRs was transcribed by Pol I. *Rluc* reporter RNA served as a template to generate *Rluc* mRNA under the promoter contained in the viral 3'-UTR in the presence of viral RdRp and NP. The pGL4.13 plasmid that constitutively expresses firefly luciferase was used as a reference to normalize the Rluc activity. The results showed that relative Rluc activity was low in 293T cells cotransfected with P1-WT or P2-WT and pcDNA-RdRp (Fig 2B). The relative Rluc activity was considerably increased by the addition of NP expression (Fig 2B) demonstrating that viral RdRp can recognize the panhandle structure formed by the 3'- and 5'-UTRs of viral RNA1 or RNA2 with the assistance of NP, and the viral 3'-UTRs had promoter activities.

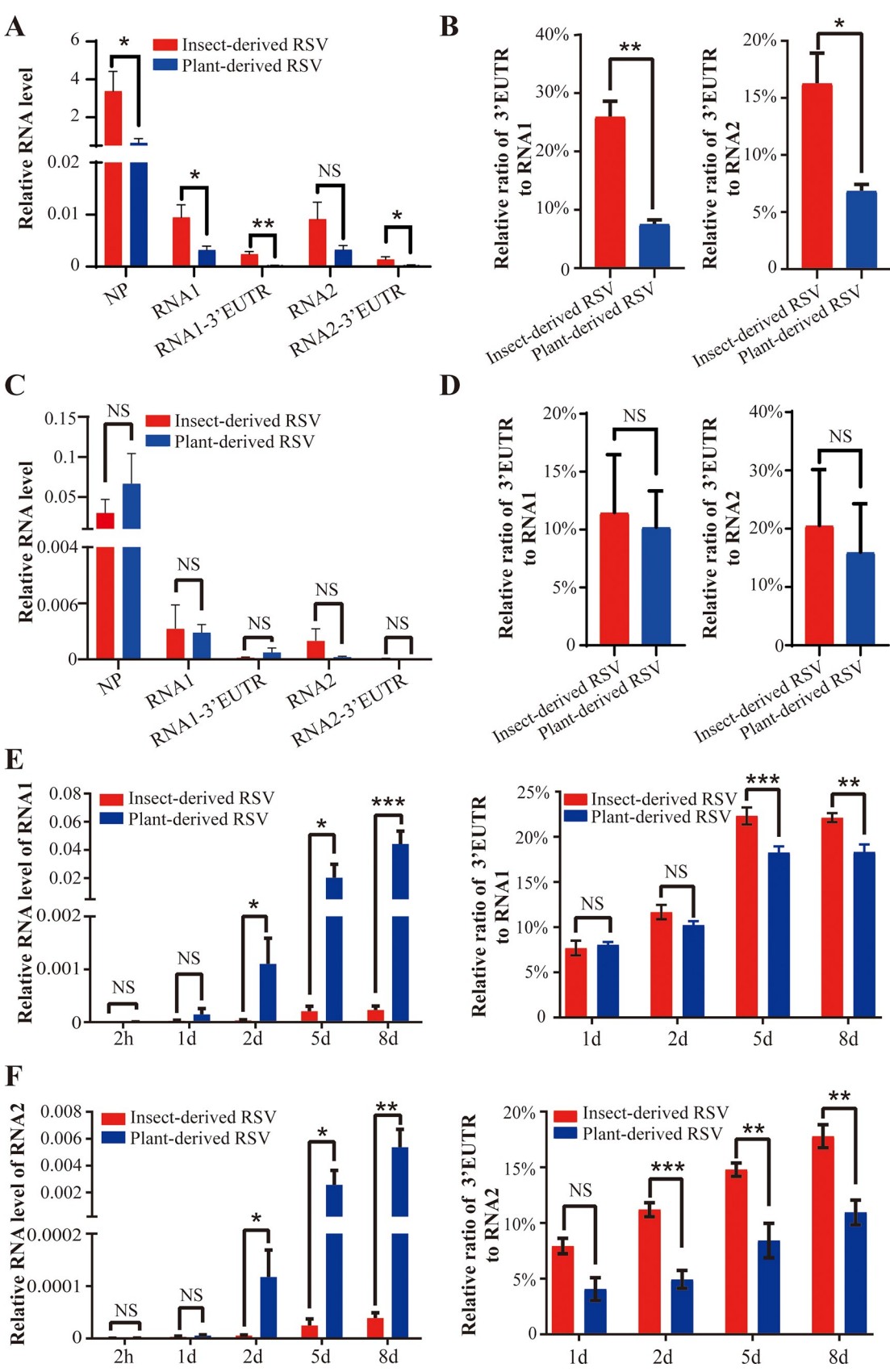

**Fig 1. The 3' extensions suppress viral replication in planthopper.** (A) Relative RNA levels of the viral *NP* gene, genomic RNA1 and RNA2, and 3' extensions of RNA1 (RNA1-3'-EUTR) and RNA2 (RNA2-3'-EUTR) in crude RSV extract from RSV-infected small brown planthopper (insect-derived RSV) and rice (plant-derived RSV). Viruliferous fourth-instar nymphs and rice leaves with obvious stripe disease symptoms were sampled for virus extraction. (B) Relative ratio of the 3' extension (3'-EUTR) to RNA1 or RNA2 in insect- and plant-derived RSVs. (C) Relative RNA levels of viral *NP*, RNA1, RNA2, RNA1-3'-EUTR, and RNA2-3'-EUTR in the diluted crude viral extracts containing equal amounts of insect- or plant-derived RSVs. (D) Relative ratio of the 3'-EUTR to RNA1 or RNA2 in the diluted crude viral extracts containing equal amounts of insect- and plant-derived RSVs. (E) Relative RNA levels of viral RNA1 and relative ratio of the 3'-EUTR to RNA1 in planthopper at various time points after feeding the artificial diet containing equal amounts of insect- or plant-derived RSVs. (F) Relative RNA levels of viral RNA2 and relative ratio of the 3'-EUTR to RNA2 in planthopper at various time points after feeding the artificial diet containing equal amounts of insect- or plant-derived RSVs. NS, no significant differences. The values are presented as the mean ± SE. *, $P < 0.05$. **, $P < 0.01$. ***, $P < 0.001$.

Then, we replaced the 3'-UTRs with the extended 3'-UTRs (3'-EUTRs) of viral RNA1 and RNA2 to construct the mutant plasmids named P1-EUTR for RNA1 and P2-EUTR for RNA2. Another type of mutant plasmids with truncated 3'-UTRs (3'-TUTRs) lacking the terminal 16 nt of viral RNA1 and RNA2 was constructed and named P1-TUTR and P2-TUTR, respectively (Fig 2A). When equal amounts of P1/P2-WT, P1/P2-EUTR, and P1/P2-TUTR were transfected into 293T cells in combination with the vectors for the expression of NP and RdRp, the cells transfected with P1-EUTR or P2-EUTR had considerably lower relative Rluc activities than those in the cells transfected with P1-WT or P2-WT and similar activities to those in the cells transfected with P1-TUTR or P2-TUTR (Fig 2B). The results of these *in vitro* experiments showed that the 3' extensions of RNA1 and RNA2 inhibit the promoter activity of viral 3'-UTRs in human cells.

## The 3' extensions inhibit viral promoter activity in planthopper

The *in vitro* results indicated that viral 3' extensions influence the promoter activity in human cells. Considering the species specificity of RNA Pol I promoters, we cloned the Pol I promoter of small brown planthopper to replace the original human Pol I promoter by querying the *L. striatellus* genome [17] based on the high conservation of Pol I promoter sequences [18,19]. The nucleotides from 15,877 to 15,906 in contig36944 have high identity with the sequences adjacent to the known Pol I transcription initiation sites of other eukaryotes, and T at position -1 and A or G at positions +1 and -2, respectively, are conserved (Fig 2C) [19]. Thus, the 370 bp sequence upstream of the planthopper Pol I transcription initiation site (from -1 to -370) (GenBank registration number MW353861) was cloned as a putative Pol I promoter.

The human Pol I promoters in the P1/P2-WT and P1/P2-EUTR plasmids were replaced by the 370 bp planthopper Pol I promoter, and *Rluc* was replaced by *GFP* for convenient observation *in vivo*. The new recombinant plasmids were named LsP1/P2-WT and LsP1/P2-EUTR (Fig 2D). When equivalent amounts of LsP1-WT or LsP2-WT were injected into viruliferous and nonviruliferous planthopper nymphs, GFP signals were clearly detected in the gut and salivary glands of viruliferous planthoppers at 6 DAI, and no signal was observed in these tissues in nonviruliferous planthoppers. Furthermore, the GFP signals were stronger in the gut of viruliferous insects than that in the salivary glands (Fig 2E). This result confirmed that the planthopper Pol I promoter was cloned and demonstrated that the promoters of the 3'-UTRs of viral RNA1 and RNA2 are active *in vivo*. Planthoppers injected with LsP1-EUTR or LsP2-EUTR had considerably weaker GFP signals in the gut at 6 DAI compared with that in viruliferous planthoppers injected with LsP1-WT or LsP2-WT (Fig 2F). The data of western blot also demonstrated lower protein expression of GFP in the gut after the injection with LsP1-EUTR or LsP2-EUTR (Fig 2G) demonstrating that the 3' extensions inhibit the promoter activity of viral RNA1 or RNA2 in planthopper. Therefore, these *in vivo* experiments

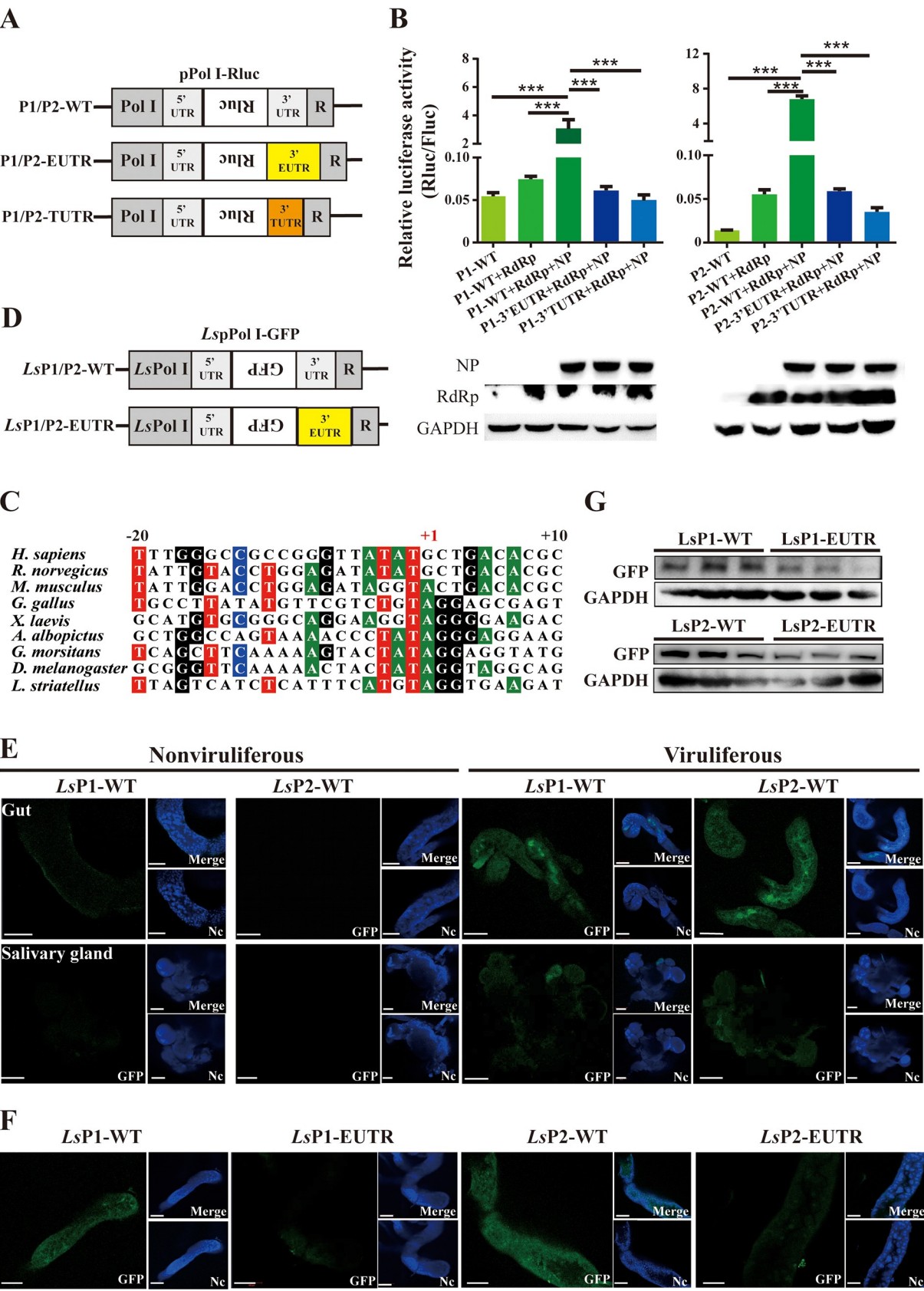

**Fig 2. The 3' extensions inhibit viral promoter activity.** (A) The diagrams of the recombinant pPol I plasmids with human Pol I promoter. The antisense *Renilla luciferase* coding sequence (Rluc) was flanked by the 5'-UTR and variable length of 3'-UTRs of RNA1 or RNA2 of RSV. Three types of the pPol I plasmids were constructed based on variable length of the 3'-UTR: P1/P2-WT with the wild type 3'-UTRs of RNA1 or RNA2; P1/P2-EUTR with the 16 or 15 nt extended 3'-UTRs of RNA1 or RNA2; and P1/P2-TUTR with the truncated 3'-UTRs lacking the terminal 16 nt regions of RNA1 or RNA2. R, gene encoding hepatitis delta virus ribozyme. (B) The promoter activity of various recombinant pPol I plasmids in human 293T cells with the coexpression of RSV *NP* and *RdRp* according to the data of the dual luciferase reporter assay. The pGL4.13 plasmid constitutively expressing firefly luciferase was transfected as a reference. The relative activity of *Renilla* luciferase (Rluc) to firefly luciferase (Fluc) is presented as the mean ± SE. ***, $P < 0.001$. The coexpression of RSV *NP* and *RdRp* encoded by the pcDNA3.1 plasmid was assayed by western blot using an in-house generated anti-NP monoclonal antibody and anti-RdRp polyclonal antibodies. GADPH was detected using anti-GADPH polyclonal antibodies and was used as an internal control. (C) Alignment of the sequences surrounding the Pol I transcription initiation sites of *Laodelphax striatellus* and other eukaryotes (*Homo sapiens*, *Rattus norvegicus*, *Mus musculus*, *Gallus gallus*, *Xenopus laevis*, *Aedes albopictus*, *Glossina morsitans*, and *Drosophila melanogaster*). The transcription initiation site is labeled as +1. (D) The diagrams of the recombinant pPol I plasmids with Pol I promoter of small brown planthopper (*Ls*Pol I). The antisense GFP coding sequence was flanked by the 5'-UTR and 3'-UTR (*Ls*P1/P2-WT) or the 5'-UTR and 3'-EUTR (*Ls*P1/P2-EUTR) of RSV RNA1 or RNA2. (E) Detection of the GFP signals in the gut and salivary glands of viruliferous planthopper 6 d after the injection of *Ls*P1-WT or *Ls*P2-WT. No signals were observed in the two organs of nonviruliferous planthoppers administered the same injection. (F) Comparison of the GFP signals in the gut of viruliferous planthopper 6 d after the injection of equal amounts of *Ls*P1-WT and *Ls*P1-EUTR or equal amounts of *Ls*P2-WT and *Ls*P2-EUTR. For (E) and (F), the nuclei (Nc) were stained with Hoechst. Scale bars: 100 μm. (G) Western blot of the GFP protein levels in the gut of (F) using an anti-GFP polyclonal antibody. GADPH was detected using anti-GADPH polyclonal antibodies and was used as an internal control.

supported the hypothesis that the 3' extensions of RNA1 and RNA2 influence the promoter activity of viral 3'-UTRs to control viral replication in planthopper.

## The extended 3'-UTR of viral RNA1 is targeted by planthopper miR-263a

To determine whether the 3' extensions of viral RNA1 and RNA2 are regulated by endogenous miRNAs in planthopper, we predicted candidate miRNAs that may target the 3' termini of viral RNA1 or RNA2 using the small RNA (sRNA) sequencing data from our previous study [20]. Two out of 213 putative miRNAs from small brown planthopper, dme-miR-263a-5p (miR-263a) and ame-miR-190 (miR-190), were predicted by both the miRanda and RNAhybrid algorithm to specifically target the 3'-EUTRs of RNA1 and RNA2, respectively. The seed sequences from 2 to 8 nt of miR-263a were a nearly perfect match to the 3'-EUTR (with a G:U wobble base pair), but not to the 3'-UTR of RNA1, and the seed sequences of miR-190 had two G:U wobble base pairs with the 3'-EUTR and did not match the 3'-UTR of RNA2 (S2 Fig). Direct interactions of these two miRNAs with the target viral RNA sequences were initially verified by dual luciferase assay in 293T cells. Relative Rluc activities in the cells transfected with the construct containing the 3'-EUTR of RNA1 were significantly decreased in the presence of miR-263a at the concentrations from 10 nM to 100 nM in a dose-dependent manner (Fig 3A). In contrast, transfection of the cells with the construct containing the 3'-UTR of RNA1 did not result in suppressed Rluc activities in the presence of miR-263a (Fig 3A). This result showed that miR-263a targeted the 3'-EUTR, but not the 3'-UTR, of RSV RNA1. Additionally, the relative Rluc activities in the cells transfected with the constructs containing 3'-EUTR or 3'-UTR of viral RNA2 were not influenced by miR-190 at the concentrations from 50 pM to 100 nM (Fig 3B) implying that these two types of 3' termini are not the targets of miR-190.

To verify the interaction between miR-263a and the 3'-EUTR of viral RNA1 *in vivo*, we performed RNA immunoprecipitation (RIP) assay using an in-house generated planthopper anti-argonaute 1 (Ago1) monoclonal antibody [21] in viruliferous planthoppers after injected with agomir of miR-263a or miR-190, which made the endogenous levels of the two miRNAs increase by 4- or 1.3-fold (Fig 3C). The 3'-EUTR of viral RNA1 was significantly enriched in the Ago1-immunoprecipitated RNAs from viruliferous planthoppers injected with agomir-263a compared to that in the control group injected with agomir-NC (Fig 3C). On the other hand, no significant enrichment in 3'-EUTR of viral RNA2 was observed in Ago1-immunoprecipitated RNAs from insects injected with agomir-190 compared to that in the control

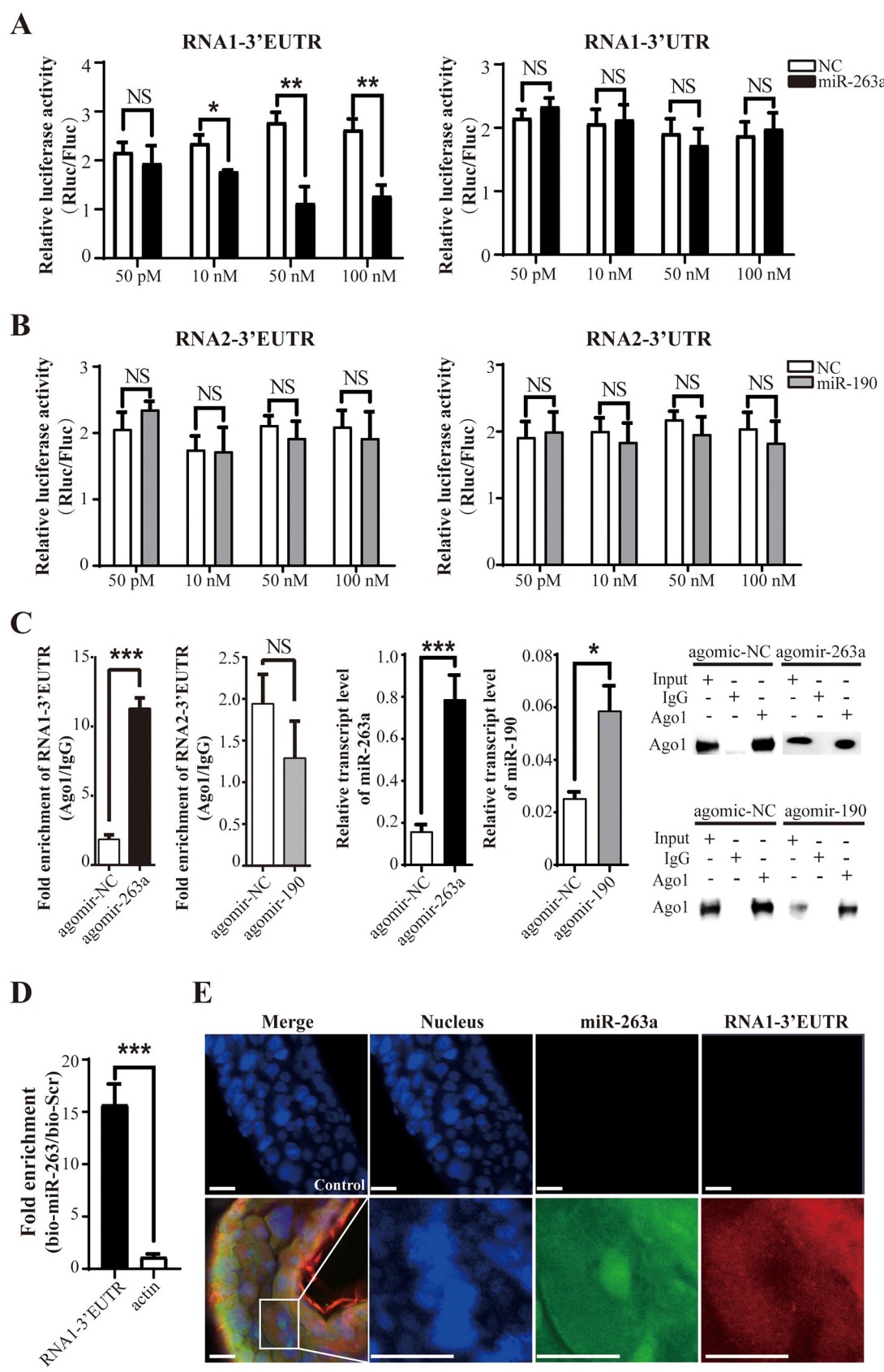

**Fig 3. The extended 3' UTR of viral RNA1 is targeted by planthopper miR-263a.** (A) Dual luciferase reporter assay in 293T cells cotransfected with miR-263a mimics and the recombinant pLightSwitch plasmid containing the 3'-untranslated terminal region (3'-UTR) or the extended 3'-UTR (3'-EUTR) of RSV RNA1 segment. (B) Dual luciferase reporter assay in 293T cells cotransfected with miR-190 mimics and the recombinant pLightSwitch plasmid containing the 3'-UTR or 3'-EUTR of RSV RNA2 segment. The mimic control (NC) was used in the control group instead of miR-263a or miR-190. The pGL4.13 plasmid expressing firefly luciferase (Fluc) was used as a reference control. The relative activity of *Renilla* luciferase (Rluc) encoded by the recombinant pLightSwitch normalized to Fluc is presented as the mean ± SE. (C) RIP analysis combined with real-time quantitative PCR (qPCR). An anti-Ago1 monoclonal antibody was used to pull down RNAs from viruliferous planthoppers injected with agomir-263a, agomir-190, or agomir control (agomir-NC). Mouse IgG was used as a negative control. The amount of 3'-EUTRs of RSV RNA1 or RNA2 in the Ago1 immunoprecipitates relative to that in the IgG samples (fold enrichment) was quantified using qPCR and compared in the groups treated with agomir-263a or agomir-190 injection and agomir-NC injection. miR-263a and miR-190 in the precipitates were assayed using qPCR. The level of Ago1 in the precipitates is shown in the western blot. NS, no significant differences. (D) Biotin pulldown assay coupled with qPCR was used to quantify the amount of 3'-EUTR of RNA1 in viruliferous planthoppers injected with biotinylated miR-263a (bio-miR-263a) or with control biotinylated scrambled sequences (bio-Scr). The transcript levels of *actin* were quantified and used as a negative control. From (A) to (D), the values are presented as the mean ± SE. NS, no significant differences. *, $P < 0.05$. **, $P < 0.01$. ***, $P < 0.001$. (E) Double fluorescence *in situ* hybridization (FISH) for miR-263a and the 3'-EUTR of RNA1 in the gut cells of viruliferous planthopper. Probes for *Caenorhabditis elegans* cel-miR-67-3p and rice *PsbP* gene were used in the control group. The probes for microRNAs were labeled with biotin. The probes for the 3'-EUTR of RNA1 and *PsbP* were labeled with digoxigenin. The nuclei were stained with Hoechst. Scale bars: 50 μm.

group (Fig 3C). Biotinylated miRNA pulldown assay confirmed the interaction between miR-263a and the 3'-EUTR of RNA1. Higher amounts of the 3'-EUTR of RNA1 were pulled down by biotinylated-miR-263a from viruliferous planthoppers than that by *actin* RNA used as a negative control (Fig 3D). Furthermore, colocalization of miR-263a and the 3'-EUTR of RNA1 in the gut cells was observed in the cytoplasm and partial region of the nucleus according to the data of double fluorescence *in situ* hybridization (FISH) (Fig 3E).

## miR-263a facilitates RSV replication by counteracting the inhibitory effect of RNA1 3' extension

Since the 3'-EUTR of viral RNA1 is targeted by miR-263a, we aimed to determine whether miR-263a regulates the influence of the 3' extension on the viral promoter activity. When P1-EUTR was cotransfected with miR-263a mimic or a control mimic (mimic-NC) into 293T cells in the presence of viral RdRp and NP, the transfection of miR-263a mimic significantly increased the normalized Rluc activity encoded by P1-EUTR compared to that in the mimic-NC-transfected group but did not influence the Rluc activity in the cells transfected with P1-WT (Fig 4A). Northern blot using a probe for *Rluc* mRNA detected the same size of the *Rluc* transcript driven by 3'-EUTR of RNA1 with or without miR-263a mimic indicating that miR-263a does not degrade the *Rluc* transcript (Fig 4B). Injection of the mixture of LsP1-EUTR and agomir-263a into viruliferous planthoppers increased the GFP expression levels in the gut compared to that in the control group injected with the mixture of LsP1-EUTR and agomir-NC according to the values of the fluorescence signals and western blot data (Fig 4C and 4D). In contrast, agomir-263a did not influence the GFP expression in the insects treated with LsP1-WT (Fig 4C and 4D). These results indicated that miR-263a counteracts the inhibitory effect of the 3' extension on the promoter activity of RNA1 3'-UTR.

To determine the role of miR-263a in RSV replication, we delivered antagomir or agomir of miR-263a into viruliferous planthoppers. The treatment with antagomir led to a 62% decrease in the expression of miR-263a, and the RNA levels of viral RNA1, *RdRp*, and *NP* in the whole body were declined by 65%, 49%, and 55%, respectively, at 4 DAI (Fig 4E). The protein level of *NP* was significantly decreased (Fig 4F). In contrast, the addition of miR-263a agomir did not upregulate the RNA levels of viral RNA1, *RdRp*, and *NP* or the protein level of *NP* in the whole body (Fig 4E and 4F) probably due to a high viral load in viruliferous

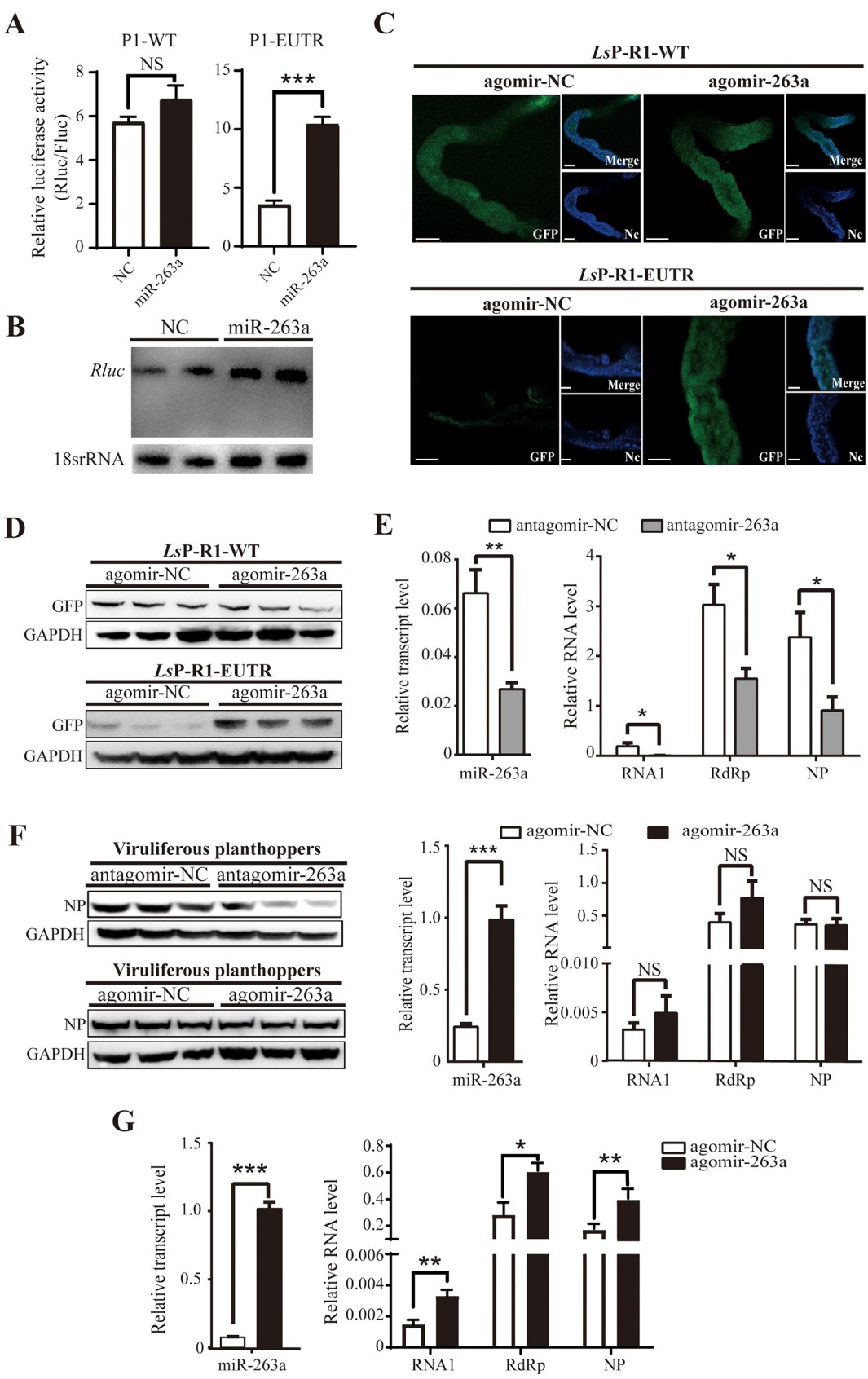

**Fig 4. miR-263a facilitates RSV replication by counteracting the inhibitory effect of RNA1 3' extension.** (A) The promoter activity of the recombinant pPol I plasmids P1-WT and P1-EUTR in the presence of miR-263a mimics or mimic control (NC) in human 293T cells was determined by dual luciferase reporter assay. P1-WT, P1-EUTR, and other experimental details are the same as described in the legend in Fig 2A and 2B. (B) Northern blot shows the size of *Rluc* transcript encoded by P1-EUTR in the presence of miR-263a mimics or NC in 293T cells; the signal was detected using a biotin-labeled LNA oligonucleotide probe for *Rluc*. 18S rRNA was used as an internal control. (C) GFP signal in the gut of viruliferous planthopper 6 d after coinjection of agomir-263a with *Ls*P1-WT or *Ls*P1-EUTR. The agomir control (agomir-NC) was injected as a negative control. *Ls*P1-WT and *Ls*P1-EUTR are described in Fig 2D. The nuclei (Nc) were stained with Hoechst. Scale bars: 100 μm. (D) The western blot results show the GFP protein levels in the gut of (C); the signal was detected using the anti-GFP polyclonal antibodies. GADPH was detected using anti-GADPH polyclonal antibodies and was used as an internal control. (E) The RNA levels of miR-263a, viral RNA1, *RdRp*, and *NP* in the whole body of viruliferous planthopper determined by real-time quantitative PCR (qPCR) 4 d after the injection of antagomir-263a or agomir-263a. The antagomir control (antagomir-NC) or agomir-NC were injected as a negative control. (F) Western blot analysis of RSV NP protein in the samples of (E) using an in-house generated anti-NP monoclonal antibody. GADPH was detected using anti-GADPH polyclonal antibodies and was used as an internal control. (G) The results of qPCR show the RNA levels of miR-263a, viral RNA1, *RdRp*, and *NP* in planthoppers infected by RSV for 5 d after the injection of agomir-263a or agomir-NC. Nonviruliferous planthoppers were initially fed RSV-infected rice for 1 d before the agomir injection. The values in (A), (E), and (G) represent the mean ± SE. NS, no significant differences. *, $P < 0.05$. **, $P < 0.01$. ***, $P < 0.001$.

planthoppers. The effects of antagomir or agomir of miR-263a on RSV load in the gut dissected from viruliferous planthoppers at 4 DAI were similar to those observed in the whole body assays (S3 Fig). To evaluate the role of miR-263a agomir in RSV replication, nonviruliferous planthoppers were initially fed RSV-infected rice for 1 d; then, the insects were injected with miR-263a agomir. The RNA levels of RNA1, *RdRp*, and *NP* were significantly upregulated by miR-263a agomir at 4 DAI (Fig 4G). Therefore, miR-263a facilitated RSV replication by eliminating the inhibitory effect of the 3' extension on the promoter activity of RNA1 3'-UTR.

## RSV suppresses miR-263a expression in planthopper

The effect of RSV infection on miR-263a expression was investigated using qPCR. No significant changes in miR-263a expression were detected in the whole body of viruliferous planthoppers compared to that in nonviruliferous planthoppers (Fig 5A). However, a significant downregulation of miR-263a expression was observed in the gut and salivary glands of viruliferous insects, and miR-263a expression was higher in the gut compared with that in the salivary glands, brain, fat body, ovary, and testis (Fig 5B). Considering that RSV proliferates mainly in the gut, the expression of miR-263a was further tested in the gut during RSV infection. After nonviruliferous planthoppers were infected with RSV for 2, 5, and 8 d by feeding RSV-containing crude extract, the expression of miR-263a in the gut was dramatically decreased concomitant to an increase in the virus load over time (Fig 5C).

To determine the reason for a reduction in miR-263a in response to RSV infection, we used the genome sequence of small brown planthopper to predict primary miR-263a (pri-miR-263a) and stem-loop precursor miR-263a (pre-miR-263a) (S4 Fig) [17]. Partial pri-miR-263a sequence of 455 bp (GenBank registration number MW353862) was able to produce a mature miR-263a transcript in 293T cells treated with various template concentrations (Fig 5D) thus validating the identity of pri-miR-263a and pre-miR-263a. Then, the transcript levels of pri-miR-263a and pre-miR-263a were measured using qPCR and compared in nonviruliferous and viruliferous gut. The results showed that pri-miR-263a and pre-miR-263a were downregulated in the gut cells (Fig 5E) and even in the nuclei of the gut cells in viruliferous planthopper (Fig 5F) where primary miRNA transcription and precursor miRNA processing occur. When double-stranded RNA of RSV *NP* (ds*NP*-RNA) and RSV crude preparations were simultaneously injected into nonviruliferous planthoppers, viral replication in terms of *NP* RNA level was restrained (S5 Fig), and the transcript levels of miR-263a and pri-miR-263a were significantly upregulated compared to those in the ds*GFP*-RNA-injected group at 5 DAI (Fig 5G).

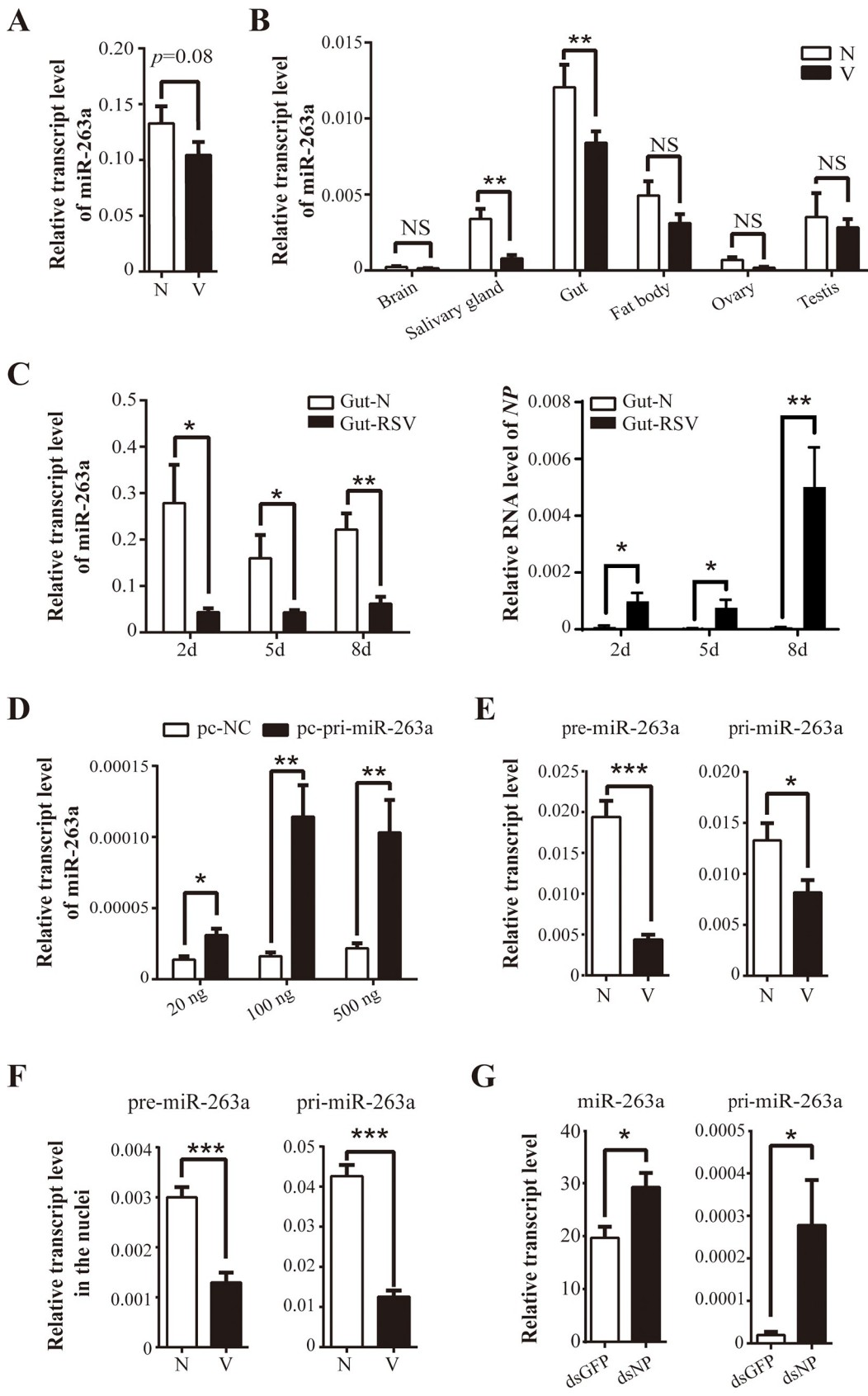

**Fig 5. RSV suppresses miR-263a expression in planthopper.** (A) Relative transcript levels of miR-263a in nonviruliferous (N) and viruliferous (V) fourth-instar nymphs of planthopper. (B) Relative transcript levels of miR-263a in six tissues of N and V planthoppers quantified by qPCR. (C) Expression patterns of miR-263a and viral *NP* in the gut of planthopper assayed using qPCR 2–8 d after the infection with RSV from RSV-infected rice. (D) Relative transcript levels of miR-263a in 293T cells assayed by qPCR after the transfection with various amounts of the recombinant pcDNA3.1 plasmid containing a partial sequence of pri-miR-263a centered pre-miR-263a (pc-pri-miR-263a). The empty pcDNA3.1 plasmid (pc-NC) was transfected and used as a negative control. (E) Relative transcript levels of pre-miR-263a and pri-miR-263a assayed by qPCR in the whole body of N and V planthoppers. (F) Relative transcript levels of pre-miR-263a and pri-miR-263a assayed by qPCR in the nuclei of N and V planthoppers. (G) Relative transcript levels of miR-263a and pri-miR-263a in the whole body of planthoppers 5 d after the injection of crude RSV preparations and double-stranded RNAs of RSV *NP* (dsNP) or double-stranded RNAs of GFP (dsGFP). The values from (A) to (G) were determined by real-time quantitative PCR and represent the mean ± SE. NS, no significant differences. [*], $P < 0.05$. [**], $P < 0.01$. [***], $P < 0.001$.

Therefore, RSV suppresses miR-263a expression by inhibiting the transcription of pri-miR-263a.

## Discussion

RSV is a typical insect-borne propagative phytovirus that circulates and replicates in insect vectors and host plants and maintains a limited replication level in insect vectors. In this study, we showed a molecular mechanism that controls the replication level of RSV in insect vectors. When planthoppers acquire virions from rice, most virions have normal genomes with complementary 3'- and 5'-UTRs. With viral replication, a 16 nt and a 15 nt sequence are added at the 3'-UTRs of RNA1 and RNA2. The 16 and 15 nt extensions in the 3' termini of the RSV genomes impair viral promoter activity resulting in the inhibition of viral replication. This inhibitory effect of the 3' extension of viral RNA1 was decreased by endogenous insect miR-263a, which targets the extended sequence. However, the expression of miR-263a was negatively regulated by RSV infection. This interaction between RSV and insect miRNAs secures the controlled replication of RSV in planthopper reflecting a distinct strategy of adaptation of phytoviruses to the insect vectors.

This study is the first to determine the promoter activity of the panhandle structure formed by RSV 3'- and 5'-UTRs in insects. Terminal complementary sequences are ubiquitous in RNA viruses, especially negative-strand RNA viruses, and play the key role in viral replication and expression [6,22]. The 3'- and 5'-termini of RSV genomic RNAs were initially detected in the RSV isolate T derived from RSV-infected maize host. The data of two-dimensional mobility shift analyses indicated that the 18–20 nt sequences at the 5'- and 3'-termini of each viral RNA are complementary to each other and generate a panhandle structure [14]. The promoter functions of the 3'- and 5'-termini of RSV were explored *in vitro* using a model RNA transcribed from a synthetic DNA template, T7 RNA polymerase, and viral 16–17 nt terminal conserved sequences of all four genomic RNA segments. Approximately 50 nt RNA products were successfully identified by incubation with solubilized viral proteins (containing at least RdRp and NP) [15]. However, the promoter activity was not verified *in vivo*. A reverse genetic system was constructed in the present study to test viral promoter activity in planthopper. This system is based on a modified pPol I plasmid with the Pol I promoter of small brown planthopper and was used to transcribe the RNA template containing viral 3'- and 5'-UTRs recognized by RSV RdRp to drive the expression of the *GFP* gene. The promoter activities corresponding to the GFP signals were detected in the gut and salivary glands of viruliferous planthopper indicating the validity of this reverse genetic system and confirming the promoter functions of the terminal panhandle structure of RSV *in vivo*.

The extensions at the 3'-UTRs of RSV inhibited viral replication. Variations in the viral UTRs generally inhibit viral replication and/or infection [7,23]. Mutagenic analysis revealed that the conserved terminal 11 nt in the terminal complementary 3'- and 5'-UTRs of Bunyamwera virus are critical for viral replication and transcription [24]. Separate mutations within

the cyclization sequences of the 5'- or 3'-UTR of Kunjin virus significantly inhibited viral replication in BHK cells, and simultaneous compensatory mutations in both cyclization sequences restored viral replication ability, suggesting that base pairing is vital for Kunjin virus replication [22]. Furthermore, the UTRs of RNA viruses account for differences in viral fitness in the host and vector cells. For example, mutations in the 3'-UTR of Chikungunya virus reduced viral replication only in the mosquito vector cells, but not in the mammalian host cells, suggesting that adaptation to the vectors rather than to the hosts is the major evolutionary force acting on viral 3'-UTR [4]. The duplicated stem loop structure of the 3'-UTR was beneficial for dengue virus replication in mammalian cells, while deleterious in mosquito cells, suggesting that viral 3'-UTR is involved in insect-host adaptation cycle [25]. The enrichment of 3'-terminally extended RSV in planthopper is good for maintaining a limited replication level for virus to strike a balance with insect immune systems without causing severe diseases. It is possible that the extension sequences are generated by viral RdRp with the template of viral genomic RNAs because the 16 nt and 15 nt extension sequences at the 3'-UTRs of RNA1 and RNA2 are respectively identical to the 8874 nt to 8889 nt fragment in the upstream region of RNA1 and the 3241 nt to 3255 nt fragment in the upstream region of RNA2 [10].

miR-263a promoted RSV replication by targeting the extended 3'-UTR of the viral genome. In contrast to the positive regulation, cellular miRNAs usually suppress viral replication or infection by direct targeting of the viral genomes. For example, human miR-32 effectively inhibited the accumulation of the retrovirus primate foamy virus type 1 [26]. Human hsa-miR-296-5p restricted enterovirus A71 replication [27]. Human miR-125-5p and miR-122 inhibited hepatitis B virus replication by targeting the coding regions of the S protein and polymerase or the 3'-UTR of the core protein [28,29]. Positive regulation of viral replication by cellular miRNAs due to direct interaction with viral genomes is unusual. For example, human miR-122 benefits hepatitis C virus replication by binding to the 5'-UTR of the viral genome to enhance viral protein translation via recruitment of the 48S ribosomal subunit [30] and to increase viral RNA accumulation by protecting the 5'-terminus of genomic RNA [31,32]. We demonstrate that miR-263a can rescue the viral promoter activity of the extended UTR, which was not apparently degraded by miR-263a. The miR-263a, possibly with the aid of Ago1, covers the extension sequences, rendering the UTRs-formed panhandle structure exposed to viral RdRp for binding.

Thus, we discovered a strategy of RSV adaptation in vector insects mediated by the interaction between terminal variation of the viral genome and endogenous insect miRNAs. This elaborate coordination limits the replication of RSV to avoid serious damage to planthopper.

## Materials and methods

### Virus, insect vector, and host plant preparation

The viruliferous small brown planthopper populations were established from a rice field population collected in Nanjing, Jiangsu Province, China. Viruliferous and nonviruliferous planthoppers were reared separately on 2–3 cm seedlings of rice (*Oryza sativa* Huangjinqing) in glass incubators as described previously [33]. To maintain the RSV-carrying frequency of the viruliferous strain of at least 90%, nonviruliferous individuals were identified and eliminated via dot enzyme-linked immunosorbent assay (dot-ELISA) using an in-house generated monoclonal anti-NP antibody every three months [33].

### Cells

Human embryonic kidney HEK293T (293T) cells were maintained in Dulbecco's modified Eagle's medium (DMEM) supplemented with 10% fetal bovine serum (FBS) at 37°C in an atmosphere of 5% $CO_2$.

## Feeding RSV crude preparations to planthoppers

Insect- or plant-derived crude RSV preparations were extracted from the RSV-infected rice leaves or viruliferous planthoppers, respectively [33]. Nonviruliferous fourth-instar planthopper nymphs were fed an artificial diet containing equal amounts of insect- or plant-derived RSV virions as described previously [33]. After feeding the diet for 24 h, the RSV-infected nymphs were transferred to healthy rice seedlings and incubated for 2 h, 1 d, 2 d, 5 d, and 8 d. All planthoppers were collected for RNA extraction and quantitative real-time PCR (qPCR) analysis.

After feeding the artificial diet containing an appropriate concentration of plant-derived RSV for 2 d, 5 d, and 8 d, the gut from the nymphs was dissected for RNA extraction and qPCR analysis.

## Double-stranded RNA synthesis and delivery

dsRNA was generated using a T7 RiboMAX express RNAi system (Promega, Madison, Wisconsin, USA) and purified using a Wizard SV gel and PCR clean-up system (Promega) following the manufacturers' protocols. A 427-bp dsRNA for viral *NP* gene was generated using the ds*NP*-F and ds*NP*-R primers (S1 Table). A 420-bp dsRNA for the green fluorescent protein (GFP) gene was amplified as a negative control (S1 Table). dsRNAs (23 nL of 6 mg mL$^{-1}$ solution) with crude preparations of insect-derived RSV were delivered into the hemolymph of nonviruliferous fourth-instar nymphs by microinjection through a glass needle using a Nano-liter 2000 system (World Precision Instruments, Sarasota, Florida, USA).

## RNA preparation and quantitative real-time PCR for the assays of miRNAs, viral RNAs, and genes

Five to eight insects, or 20 salivary glands, 10 brains, 8 guts, and 8 fat bodies from nonviruliferous or viruliferous fourth-instar planthopper nymphs, or 8 samples of ovaries and testes from nonviruliferous or viruliferous planthopper adults, or RIP pulldown materials were used for RNA extraction by Trizol reagent (Invitrogen). DNase (Promega) was added to eliminate DNA contamination in the RNA samples. After quality control with a NanoDrop spectrophotometer (Thermo Scientific, Waltham, MA, USA) and gel electrophoresis, 2 μg of total RNA was reverse transcribed to cDNA in 20 μL by using a Superscript III first-strand synthesis system (Invitrogen) and random primers (Promega, Madison, WI, USA) according to the manufacturer's instructions [33]. For reverse transcription of miRNAs, 1 μg of RNA was reverse transcribed by using miRNA RT enzyme mix of a miRcute plus miRNA first-strand cDNA kit (Tiangen, Beijing, China) [34].

The relative expression levels of miRNAs, planthopper genes, or viral RNAs were quantified by a miRcute miRNA qPCR detection kit (Tiangen) or LightCycler 480 SYBR Green I Master (Roche, Basel, Switzerland) using a LightCycler 480 instrument (Roche), respectively. Eight biological replicates were used for statistical analysis. U6 snRNA and planthopper *EF2 gene* [33] were used as endogenous controls for miRNAs, planthopper mRNAs, or viral RNAs, respectively. Viral RNA1 and RNA2 with the 3'-EUTR were identified by amplifying the 3'-UTR containing the extended terminal sequences. The fragments from the inner regions of RNA1 and RNA2 were amplified to measure the total RNA levels of RNA1 and RNA2, respectively. The ratios of RNA1 or RNA2 to the 3' EUTRs were determined by associating the relative RNA levels of the 3' EUTRs to those of the RNA levels of RNA1 and RNA2. Primers are listed in S1 Table.

To selectively quantify pri-miR-263a, pre-miR-263a, and miR-263a, specific primers were designed based on previous publications [34]. Briefly, the cDNA samples reverse transcribed

by a Superscript III first-strand synthesis system (Invitrogen) were used to quantify pri-miR-263a. The cDNA samples reverse transcribed by a miRcute miRNA qPCR detection kit (Tiangen) were used to quantify pre-miR-263a and mature miR-263a. All products of PCR amplification were sequenced to verify the specificity of the primers. Differences were statistically evaluated using Student's t-test to compare two means using SPSS 17.0. Primers are listed in S1 Table.

## Determination of the viral promoter activity *in vitro*

To determine the promoter activities of viral 3'-UTRs, the pPol I plasmid (a gift from the laboratory of Prof. Yi Shi) containing the promoter sequence of human RNA polymerase I (Pol I) was used as a backbone to construct the recombinant plasmids. The reverse complementary sequence of *Renilla luciferase* gene flanked by the 5'-UTR and variable length of 3'-UTRs of RNA1 or RNA2 of RSV was inserted into the pPol I plasmid at the BglII site to generate the P1/P2-WT recombinant plasmids. The 3'-UTRs were replaced by 3'-EUTRs to generate the P1/P2-EUTR plasmids, respectively. Two deletion mutants P1/P2-TUTR lacking the terminal 16 nt of 3'-UTR of RNA1 or RNA2 were constructed based on the P1/P2-WT plasmids, respectively, using a KOD-Plus mutagenesis kit (Toyobo Bio-Technology, Osaka, Japan). The full-length ORFs of viral *NP* and *RdRp* were cloned from viruliferous planthopper cDNA by KOD FX DNA polymerase (Toyobo Bio-Technology) and were inserted into the pcDNA3.1 vector (Invitrogen) between the NotI and XbaI or BamHI and NotI restriction sites, respectively (S1 Table). The pGL4.13 plasmid (Promega) constitutively expressing firefly luciferase was used as a reference. A total of 0.5 μg of various recombinant pPol I plasmids (including P1/P2-WT, P1/P2-EUTR, and P1/P2-TUTR), 1 μg of pcDNA3.1-RdRp, 1 μg of pcDNA3.1-NP, and 0.1 μg of the pGL4.13 vector were cotransfected into 293T cells by Lipofectamine 3000 (Invitrogen). After transfection at 37°C for 45 h, the cells were harvested for dual luciferase reporter assay performed by using a Dual-Glo luciferase assay system (Promega) and a luminometer (Promega). *Renilla* luciferase expression was normalized to firefly luciferase expression. Six independent replicates were assayed for each treatment. Differences were statistically evaluated using Student's t-test to compare two means using SPSS 17.0. Primers used in this experiment are listed in S1 Table.

## Bioinformatics analysis of RNA polymerase I promoter of planthopper

In eukaryotes, the RNA Pol I promoter is generally located in the intergenic spacers between the 28S and 18S rRNA coding sequences [17], and the transcription initiation site is usually present within a region from 3,000 to 5,000 bp upstream of the 18S rRNA gene [18]. Thus, the 5,000 bp sequence upstream of the planthopper 18S rRNA gene was identified in contig36944 by querying the *L. striatellus* genome [17] with the 18S rRNA DNA sequence (GenBank accession no. AY591581.1). Possible RNA Pol I transcription initiation site was predicted by homology search with Bioedit software (version 7.2.5) based on highly conserved nucleotides around the transcription initiation site of 18S rRNA in eukaryotes [18]. The base frequency adjacent to the Pol I transcription initiation sites of *L. striatellus* and other eukaryotes (*Homo sapiens*, *Rattus norvegicus*, *Mus musculus*, *Gallus gallus*, *Xenopus laevis*, *Aedes albopictus*, *Glossina morsitans*, and *Drosophila melanogaster*) was identified using the WebLOGO tool.

## Functional analysis of the effects of the 3'-extensions on viral promoter activities *in vivo*

A sequence ranging from -1 to -370 bp of the predicted planthopper RNA Pol I promoter was cloned using the *Ls*Pol I-F/*L*-Pol I-R primers (S1 Table) and was used to replace the original

human Pol I promoter of the P1-EUTR or P2-EUTR vectors, respectively. Additionally, the *RLuc* gene was replaced by the *GFP* gene to generate the *Ls*P1-EUTR or *Ls*P2-EUTR plasmids (5 μg; Genewiz), respectively. Then, *Ls*P1-WT or *Ls*P2-WT with normal 3'-UTR of RNA1 or RNA2 were generated by using a KOD-Plus mutagenesis kit (Toyobo Bio-Technology), respectively (S1 Table). The *Ls*P1/P2-WT plasmids (23 nL of 500 ng μL$^{-1}$ solution) were injected into nonviruliferous or viruliferous fourth-instar nymphs. Then, equal amounts of the *Ls*P1-WT and *Ls*P1-EUTR or *Ls*P2-WT and *Ls*P2-EUTR plasmids (23 nL, 500 ng μL$^{-1}$) were simultaneously injected into viruliferous fourth-instar nymphs. Six days after the injection, the nymphs were collected for protein extraction and tissue dissection. GFP protein level in the gut was evaluated by western blot using an anti-GFP monoclonal antibody (Abcam, Cambridge, UK). GADPH was detected by anti-GADPH polyclonal antibodies (Abcam) and was used as an internal control. The gut and salivary glands were dissected for GFP signal estimation performed using a Leica TCS SP5 confocal microscope (Leica Microsystems, Solms, Germany).

## microRNA analysis and prediction of a potential target in planthopper

The small RNA sequencing data of viruliferous planthopper were obtained in our previous study [20]. To generate the planthopper miRNA datasets, clean reads mapped to rRNAs, tRNAs, snoRNAs, snRNAs, or virus-derived sRNAs identified in our previous studies [17,20] with perfect matching were discarded. The remaining sRNA reads were mapped to the planthopper genome with perfect matching allowing up to one mismatch using the mapper module of the miRDeep2 (version 2.0.0.5) software package. The statistics of the read positions (discrete manner) and frequency of reads within a potential hairpin are scored as measurement of the posterior probability that the hairpin is the stem-loop of planthopper miRNAs. The quantifier module was used to evaluate the expression levels based on read counts.

To determine whether 3' extensions altered viral RNA-miRNA interactions, the miRanda [35] and RNAhybrid [36] algorithms were used to predict the potential binding sites of planthopper miRNAs in the 3' extensions within approximately 200 bp upstream flanks of the RSV genome. The minimum free energy (MFE) of the RNA duplex was analyzed by miRanda using a cutoff of -18 kcal mol$^{-1}$. The parameters of RNAhybrid were set as 1 hits per target, free energy threshold of –20 kcal mol$^{-1}$, helix constraint from 2 to 8, max bulge loop length of 2, max internal loop length of 8, and two G:U included in the seed.

## miRNA target validation

The putative targets of miR-263a (101 bp sequence containing the 3'extension of viral RNA1 and the upstream flanks) or the putative targets of miR-190 (200 bp sequence containing 15 bp of the 3' extension of viral RNA2 and the upstream flanks) were cloned and inserted into the luciferase reporter vector (pLightSwitch UTR plasmid; Active Motif, Carlsbad, CA, USA) between the BglII and NcoI restriction sites to generate pLightSwitch-RNA1-3'EUTR and pLightSwitch-RNA2-3'EUTR using the pLS-R1-EUTR-F/R and pLS-R2-EUTR-F/R primers, respectively (S1 Table). Sequences containing the 3'-UTRs of RNA1 or RNA2 were also inserted to generate pLightSwitch-RNA1-3'UTR and pLightSwitch-RNA2-3'UTR, respectively, by using a KOD-Plus mutagenesis kit (Toyobo Bio-Technology) using the pLS-R1-UTR-F/R and pLS-R2-UTR-F/R primers (S1 Table). The pGL4.13 plasmid expressing firefly luciferase (Fluc) was used as a reference control. The sequence of a *Caenorhabditis elegans* miRNA, cel-miR-67-3p (5'-UCACAACCUCCUAGAAAGAGUAGA-3'), was used as a control mimic (NC). 293T cells were cotransfected with the recombinant pLightSwitch plasmids (1 μg), miRNA/control mimics (RiboBio, China) at various concentrations (50 pM, 10 nM, 50

nM, and 100 nM), and 1 μg of the pGL4.13 plasmid using Lipofectamine 3000 (Invitrogen). miRNA mimics corresponded to the sequences of miR-263a (5'-AATGGCACTGGAAGA ATTCACGGG-3') and miR-190 (5'-AGATATGTTTGATATTCTTGG-3'). The relative activity of *Renilla* luciferase (Rluc) encoded by the recombinant pLightSwitch plasmids normalized to Fluc activity is presented as the mean ± SE. The activities of firefly and *Renilla* luciferases were measured 45 h after the transfection using a Dual-Glo luciferase assay system (Promega, Madison, USA).

### miRNA antagomir and agomir injection

miRNA inhibition or overexpression was performed by the injection of antagomir or agomir into the insects [34]. The sequence of cel-miR-67-3p was used as a negative control of antagomir or agomir (antagomir-NC or agomir-NC). A total volume of 13.8 nL of antagomir-263a/NC, agomir-263a/NC, antagomir-190/NC, or agomir-190/NC (500 μM; RiboBio) was delivered into the fourth-instar viruliferous nymphs by microinjection using a Nanoliter 2000 system (World Precision Instruments). Four days after the injection, planthoppers were collected for RNA or protein isolation.

Nonviruliferous fourth-instar nymphs were fed an artificial diet containing crude ricederived RSV extract as described previously. After feeding RSV for 24 h, the RSV-infected nymphs were injected with agomir-263a or agomir-NC, transferred to healthy rice plants, and incubated for additional 1 d and 4 d for collection. Eight replicates and five insects per replicate at each time point were prepared. Samples from this experiment were used for RNA extraction and qPCR analysis.

### RNA immunoprecipitation (RIP) assays

The RIP assay was performed using a Magna RIP RNA-binding protein immunoprecipitation kit (Merck Millpore, Billerica, MA, USA) [21]. Four days after the injection with agomir-263a/NC or agomir-190/NC, 20 fourth-instar viruliferous nymphs were homogenized in ice-cold RIP lysis buffer; the homogenate was incubated with 20 μL of magnetic beads and 2 μg of a monoclonal Ago1 antibody [21] or normal mouse IgG (Abcam) at 4˚C overnight. A 1/10 fraction of the supernatant of the lysate was stored as an "input" sample. The immunoprecipitates were digested with proteinase K to remove proteins and release RNAs. Total RNA was extracted by TRIzol reagent and reverse transcribed into cDNA using random primers. qPCR was performed to detect the expression levels of the target fragments. The RNA level of each target segment relative to that in the IgG control sample is reported as the mean ± SE. A value of Ago/IgG above one indicates that the target segment may interact with the corresponding Ago protein. Student's t-test was performed to evaluate the differences between the two means using SPSS 17.0.

### Pulldown assay with biotinylated miRNA

Biotinylated miR-263a (bio-miR-263a; 23 nL) or control biotinylated scrambled sequences (bio-Scr) (50 nM; RiboBio) was injected into the fourth-instar viruliferous nymphs by microinjection using a Nanoliter 2000 system (World Precision Instruments). Two days after the injection, the nymphs were collected, homogenized, and lysed with ice-cold lysis buffer (0.02 M Tris-HCl, 0.1 M KCl, 5 mM MgCl$_2$, and 0.5% NP-40, pH 7.5) supplemented with a proteinase inhibitor (Sigma, Santa Clara, CA, USA) and an RNase inhibitor (Promega) for 10 min. The lysates were centrifuged at 13,000 rpm for 10 min at 4˚C, and 1/10 of the supernatant was saved as an "input". To exclude RNA and protein complexes, M-280 streptavidin beads (Thermo Scientific) were blocked in lysis buffer containing RNase-free bovine serum albumin

(BSA) and yeast tRNA (Sigma). The remaining supernatant of the lysate was incubated with the beads at 4˚C for 4 h and sequentially washed with lysis buffer, low salt buffer, and high salt buffer. Bound RNAs were isolated using TRIzol reagent. RNA from biotin-labeled samples (20 μL) and 1 μg of RNA from the "input" samples were used for qPCR. Four replicates were performed. Student's t-test was performed to evaluate the differences between the two means using SPSS 17.0, and P values < 0.05 were considered statistically significant. The miRNA enrichment was calculated as follows:

$$\text{Enrichment ratio} = \frac{bio - miR\ pull\ down/bio - Src\ pull\ down}{bio - miR\ input/bio - Src\ input} \tag{1}$$

### Fluorescence in situ hybridization

Double fluorescence *in situ* hybridization (dFISH) was performed in the gut of viruliferous planthopper. Prehybridization solution contained biotin-labeled miR-263a probe (1 μmol mL$^{-1}$; RiboBio) and digoxigenin (DIG)-labeled RNA1 3'-EUTR probe (50 ng μL$^{-1}$) for hybridization. Approximately 200 bp of the RNA probe for 3'-EUTR of RNA1 was synthesized by a T7/SP6 RNA transcription kit (Roche). The gut was dissected from viruliferous fourth-instar nymphs and fixed in 4% (wt vol$^{-1}$) paraformaldehyde at 4˚C overnight. After washing with PBST, the gut was digested with proteinase K (20 μg mL$^{-1}$; Tiangen) at 37˚C for 15 min and hybridized with the miRNA probe and viral RNA probe at 37˚C overnight. The midgut was washed with 0.2× SSC at 37˚C to remove nonspecific binding. An anti-DIG alkaline phosphatase-conjugated antibody (1:500) and an anti-biotin antibody (1:100) were used for signal detection. Probes for cel-miR-67-3p and rice *PsbP* gene (KF460579) were used in the control group. The fluorescent signal of DIG or biotin was generated by HNPP/Fast Red or fluorescein-tyramide (Perkin-Elmer, Waltham, MA, USA) and recorded using a Leica TCS SP5 confocal microscope (Leica Microsystems). Primers used for probe synthesis are listed in S1 Table.

### Functional analysis of the effect of miR-263a on viral promoter activities *in vitro* and *in vivo*

To determine the effect of miR-263a on viral promoter activities *in vitro*, 80 ng of miR-263a/NC mimics were cotransfected with 0.5 μg of the P1-EUTR or P1-WT plasmids into 293T cells with coexpression of RSV *RdRp* and *NP*. Forty-five hours after the transfection at 37˚C, the cells were harvested for dual luciferase reporter assay.

To determine the effect of miR-263a on viral promoter activities in planthopper, equal amounts of *Ls*P1-WT and *Ls*P1-EUTR (23 nL, 500 ng μL$^{-1}$) were microinjected into viruliferous fourth-instar nymphs by microinjection using a Nanoliter 2000 system (World Precision Instruments). Two days after the injection, a total volume of 13.8 nL (500 μM) of agomir-263a/agomir-NC was delivered into the nymphs. Four days after the additional injection, the gut and salivary glands were dissected for GFP detection.

### Western blot

Proteins from 0.5 mL of 293T cells (1 × 10$^6$ cells mL$^{-1}$) or 20 guts or 10 bodies of the treated fourth-instar planthopper nymphs were extracted using 1×PBS (pH 7.2) at 4˚C, separated by SDS-PAGE, and detected by using anti-GADPH polyclonal antibodies (Abcam), an in-house generated anti-NP monoclonal antibody, in-house generated anti-RdRp polyclonal antibodies, or anti-*Renilla* luciferase polyclonal antibodies (Abcam). Signal of the immunoblot was detected by an eECL western blot kit (CWBIO, Beijing, China).

## Northern blot

Total RNA was extracted from 293T cells after cotransfection with the equivalent amounts of P1-EUTR, pcDNA-NP, pcDNA-RdRp, and miR-263a mimics/NC. RNA molecules were resolved by electrophoresis through a 0.9% agarose denaturing gel and transferred onto a positively charged nylon membrane (Invitrogen). Biotin-labeled LNA oligonucleotide probes (20 ng mL$^{-1}$, RiboBio) for the antisense sequence of *Rluc* mRNA (5'-bio-CCTACGAGCACCAA GACAAGATCAAGGCCA-bio-3') and 18S rRNA (5'-bio-CGGAACTACGACGGTATCTG-bio-3') were hybridized to the membrane in prehybridization/hybridization buffer (Invitrogen) at 42˚C for 16 h. The membranes were blocked with blocking buffer (Invitrogen) for 15 min at room temperature followed by incubation with hybridization buffer (Invitrogen) containing a streptavidin-HRP conjugate (1:500, Invitrogen) for 1 h at RT. Detection was carried out by an eECL western blot kit (CWBIO).

## Overexpression of miR-263a in 293T cells

A miRNA-expressing vector based on the pcDNA 3.1 expression vector (Invitrogen) was constructed to overexpress miR-263a in 293T cells. Approximately 500 bp fragments centered around pre-miR-263a were amplified from the planthopper genome and ligated into the pcDNA 3.1 plasmid between the KpnI and XhoI sites. The empty pcDNA 3.1 plasmid was used as a negative control. miR-263a expression was performed by transfection of the recombinant plasmids into 293T cells using Lipofectamine 3000. miR-263a overexpression efficiency was verified by qPCR using the primers listed in S1 Table.

## Supporting information

**S1 Fig. The secondary structure of RSV genomic RNA1 (A) and RNA2 (B) with the regular 3' untranslated terminal region (3'-UTR) or extended untranslated terminal region (3'-EUTR).** The complementary 3' and 5' termini are marked with red boxes and enlarged on the right side.
(TIF)

**S2 Fig. Sequence alignments of miR-263a (A) and miR-190 (B) with the predicted target site in the 3' terminus of RSV RNA1 or RNA2.** The 3' extensions are shown in red. The G:U wobble pairs are labeled.
(TIF)

**S3 Fig. The effects of antagomir and agomir of miR-263a on RSV replication in the gut of viruliferous planthopper.** (A) Relative transcript levels of miR-263a in the gut of viruliferous planthopper determined by real-time quantitative PCR (qPCR) 4 d after the injection of antagomir-263a or agomir-263a. Antagomir control (antagomir-NC) or agomir control (agomir-NC) were injected and used as the negative controls. (B) Relative RNA levels of RSV RNA1, *RdRp*, and *NP* in the gut of viruliferous planthopper measured by qPCR 4 d after the injection of antagomir-263a or antagomir-NC. (C) Relative RNA levels of RSV RNA1, *RdRp*, and *NP* in the gut of viruliferous planthopper measured by qPCR 4 d after the injection of agomir-263a or agomir-NC. The values in (A), (B), and (C) represent the mean ± SE. NS, no significant differences. $^{*}$, $P < 0.05$. $^{***}$, $P < 0.001$. (D) Western blot analysis of RSV NP protein in the samples of (B) and (C) using an in-house generated anti-NP monoclonal antibody. GADPH was detected using anti-GADPH polyclonal antibodies and was used as an internal control.
(TIF)

**S4 Fig. Predicted stem-loop structure of pre-miR-263a.** Mature miR-263a is marked with a line. Base-pair probabilities (from 0 to 1) are shown in the heatmap.
(TIF)

**S5 Fig. Relative RNA levels of RSV *NP* in the whole body of planthopper 5 d after the injection of crude RSV preparations and double-stranded RNAs of RSV NP (dsNP) or double-stranded RNAs of GFP (dsGFP).** The values were determined by real-time quantitative PCR and are presented as the mean ± SE. *, $P < 0.05$.
(TIF)

**S1 Table. Primers used in this study.**
(DOCX)

## Acknowledgments

We are grateful to Prof. Yi Shi from Institute of Microbiology, Chinese Academy of Sciences for providing the pPol I plasmid and guiding us to construct the reverse genetic system.

## Author Contributions

**Conceptualization:** Wan Zhao, Feng Cui.

**Data curation:** Feng Jiang.

**Funding acquisition:** Wan Zhao, Wei Wang, Feng Cui.

**Methodology:** Wan Zhao, Jinting Yu, Wei Wang.

**Project administration:** Feng Cui.

**Writing – original draft:** Wan Zhao.

**Writing – review & editing:** Le Kang, Feng Cui.

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
