## [Decision Letter · Decision Letter 0]

12 Feb 2021

Dear Dr. Cui,

Thank you very much for submitting your manuscript "Coordination between terminal variation of the viral genome and insect microRNAs regulates rice stripe virus replication in vector insects" for consideration at PLOS Pathogens. As with all papers reviewed by the journal, your manuscript was reviewed by members of the editorial board and by several independent reviewers. The reviewers appreciated the attention to an important topic. Based on the reviews, we are likely to accept this manuscript for publication, providing that you modify the manuscript according to the review recommendations.

Sincerely,

Aiming Wang, Ph.D

Associate Editor

PLOS Pathogens

Shou-Wei Ding

Section Editor

PLOS Pathogens

Kasturi Haldar

Editor-in-Chief

PLOS Pathogens

orcid.org/0000-0001-5065-158X

Michael Malim

Editor-in-Chief

PLOS Pathogens

orcid.org/0000-0002-7699-2064

Reviewer Comments (if any, and for reference):

Reviewer's Responses to Questions

**Part I - Summary**

Reviewer #1: This manuscript involves investigating the role of 3’ RNA extensions in the RNA1 and RNA2 genome segments of Rice stripe virus. The extensions were shown to inhibit virus levels and to act at the virus genome replication stage. The extensions were more prevalent in the planthopper vector, and a planthopper miRNA interacts with the RNA1 extension and reduces its inhibitory effect. In general, the experiments are well done and the data convincing. Suggestions for improvement are provided.

Reviewer #2: The manuscript submitted by Zhao et al. entitled “Coordination between terminal variation of the viral genome and 2 insect microRNAs regulates rice stripe virus replication in vector insects” describe an interesting story that the interaction of rice stripe virus (RSV, a member of tenuivirus) with it’s insect vector host small brown hopper was elaborately regulated. By using a reverse genetic system for RSV in human cells and a vector insect, they demonstrated that the 3'-terminal extensions of RSV RNA 1 & 2 suppress viral replication in vector insects by inhibiting promoter activity due to the formation of a panhandle structure by viral 3’- UTRs. They further showed that the extension sequence in the viral RNA1 segment was targeted by an endogenous insect microRNA, miR31 263a, which compromised the inhibitory effect of the extension sequence on viral promoter activity. However, the expression of miR-263a was down-regulated by RSV infection. The research is well designed and results are convincing. But before acceptance for publication, there are some issues needed to be addressed.

Reviewer #3: Zhao et al investigated the effect of 3’UTR extensions of RNA1 and RNA2 segments of rice stripe virus in viral replication in the planthopper Laodelphax striatellus. They first showed that the extensions are mainly produced when the virus replicates in planthoppers compared to when it replicates in the rice plant. The extensions were shown to inhibit replication of RSV. The authors went through extensive experimentation to show that the 3’UTRs of RNA1 and RNA2 serve as promoters and the extensions in the 3’UTRs interfere with their promoter activities by disrupting the panhandle structure normally formed by viral 3’ and 5’UTRs. Next, they found that a planthopper miRNA, miR-263a, specifically binds to the extended 3’UTR of RNA1 and decreases the inhibitory effect of the extension sequences. RSV infection itself reduced the transcription of the primary transcript of miR-263a and hence the abundance of the mature miRNA, a mechanism for regulating virus replication in planthoppers. This is a comprehensive and well executed work, with conclusions supported by the results. I only have minor comments.

**Part II – Major Issues: Key Experiments Required for Acceptance**

Reviewer #1: Line 121 – need to provide rationale/explanation here for why using Human cells to study plant/insect virus

Lines 156-169 – the details of acquiring the planthopper Pol I sequence is distracting and is not needed here – suggest moving it to M&M section

Line 223 and Fig.3E – in the FISH experiment, how would the miR-263a probe (which is complementary to miR-263a) hybridize with mirR-263a when it is base-paired to the 3’ extension?

The Discussion needs improvement:

Provide an integrate findings into working model – cycling from plant to insect.

Gamarnik lab work on Dengue’s 3’UTR in human vs mosquito cells should be discussed.

Do the extensions also inhibit replication in plant cells?

Possible mechanisms for how the 3’-extension is added should be mentioned.

Ideas on how the 3’UTR extension are preferentially added or maintained in insect cells.

Reviewer #2: Major:

1. Authors should provide detailed information of RSV population before and after given planthopper. The information includes sequences 3’ – terminal of RNA 1 & 2 and percentage of major groups such, no extension, with extensions.

2. In all of the results presented, RNA or 3’-EUTR level was only detected by Q-RT-PCR. But did not provide how the results were calculated. The authors should use other methods such sequencing or northern blots to confirm their results.

Reviewer #3: none

**Part III – Minor Issues: Editorial and Data Presentation Modifications**

Reviewer #1: Abstract – line29 change to “…due to structural interference with the panhandle structure formed…”

Line 73 – change to “.., negative-sense RNA1 and ambisense RNA2, of

Line 73 – change semicolon to a period

Figure1C – red bar for NP should have white gap where the scale changes in the y-axis

Line 119 – change “damage” to “modify”

Line 122 – change to “the human RNA polymerase I (Pol I) normally directs the synthesis of ribosomal RNAs (rRNA) [16]. – delete “that function as an RNA template”.

Line 131 – change “, rRNA of…” to “, the Rluc reporter RNA……..was transcribed by Pol I.”

Line 132 – change “Rluc rRNA served…” to “Rluc reporter RNA served..”

Line 151 – change to “inhibit the promoter activity of viral 3’-UTRs in human cells.”

Line 155 – change to “activity in human cells”

Line 195 – please indicate the criteria used to select candidate miRNAs

Fig3 – please comment on the endogenous levels of miR-263a vs the concentrations being injected

For Figs3, 4 and 5 – might want to consider standardizing the color scheme of the graphs – Figs 3 and 5 are in black & white, whereas Fig 4 is in color.

Line 353 – please expand more specifically on how the miR interaction and Ago binding could “relieve” the inhibitory effect of the 3’-extension

Reviewer #2: Minor:

1. In Fig. 1 A to D, the sampling time should be provided either in results or fig legends.

2. In Fig. 2 B, how much of P1-WT and P2-WT were added in these two experiments?

3. Line 35, rice strip virus is member of Tenuivirus genus, not Phytoreovirus.

4. Line 41 “rice strip viruses” should be corrected to “rice strip virus”.

5. Line 77 “an RdRp” should be “a RdRp”.

Reviewer #3: Line 2 and the rest of the manuscript: I believe “insect vectors” is more commonly used than “vector insects”. The latter doesn’t read as well as the former. In a small number of places in the manuscript “insect vectors” has been used. It would be good to have consistency, either way.

Line 27: insert the scientific name of the planthopper as it is not mentioned in the title or the abstract.

Line 56: one of the major…

Line 100 and 109: technically, the amount of RNA is not equivalent to the number of virions. Virion should be replaced with viral RNAs

Line 139: with the assistance of….

Line 230: the authors should be consistent with the usage of mimic/agomir. There are other instances of interchangeable usage of the terms.

Line 285: injected into…

Line 456 and 583: Functional analysis of …..

Fig. 3E: The figure is not really clear in that it doesn’t have a defined nucleus and cytoplasmic region. Does the enlarged image in the lower row represent one cell?

PLOS authors have the option to publish the peer review history of their article (what does this mean?). If published, this will include your full peer review and any attached files.

Reviewer #1: No

Reviewer #2: No

Reviewer #3: No
---

## [Editor Report · Decision Letter 1]

25 Feb 2021

Dear Dr. Cui,

We are pleased to inform you that your manuscript 'Coordination between terminal variation of the viral genome and insect microRNAs regulates rice stripe virus replication in insect vectors' has been provisionally accepted for publication in PLOS Pathogens.

Best regards,

Aiming Wang

Associate Editor

PLOS Pathogens

Shou-Wei Ding

Section Editor

PLOS Pathogens

Kasturi Haldar

Editor-in-Chief

PLOS Pathogens

orcid.org/0000-0001-5065-158X

Michael Malim

Editor-in-Chief

PLOS Pathogens

orcid.org/0000-0002-7699-2064
---

## [Editor Report · Acceptance letter]

5 Mar 2021

Dear Dr. Cui,

We are delighted to inform you that your manuscript, "Coordination between terminal variation of the viral genome and insect microRNAs regulates rice stripe virus replication in insect vectors," has been formally accepted for publication in PLOS Pathogens.

Best regards,

Kasturi Haldar

Editor-in-Chief

PLOS Pathogens

orcid.org/0000-0001-5065-158X

Michael Malim

Editor-in-Chief

PLOS Pathogens

orcid.org/0000-0002-7699-2064